# Nutrigenomics and Nutrigenetics Research in New Zealand, and Its Relevance and Application to Gastrointestinal Health

**DOI:** 10.3390/nu14091743

**Published:** 2022-04-22

**Authors:** Lynnette Ferguson, Matthew Barnett

**Affiliations:** 1Discipline of Nutrition and Dietetics, Faculty of Medical and Health Sciences, University of Auckland, Private Bag 92019, Auckland 1142, New Zealand; 2Physiology & Health Team, AgResearch Limited, Palmerston North 4442, New Zealand; matthew.barnett@agresearch.co.nz; 3The Riddet Institute, Palmerston North 4442, New Zealand

**Keywords:** nutrigenetics, nutrigenomics, transcriptomics, metabolomics, proteomics, microbiome

## Abstract

Nutrigenomics New Zealand (NuNZ) was a collaborative research programme built among three organisations—the University of Auckland, AgResearch Limited and Plant & Food Research. The programme ran for ten years, between 2004 and 2014, and was tasked with developing the then emerging field of nutrigenomics, investigating its applications to New Zealand, and potential benefits to the plant food and agricultural sectors. Since the beginning of the programme, nutrigenomics was divided into two fields—nutrigenetics and nutrigenomics. The first of these is now more commonly called personalised nutrition, and has recently been recognised and criticised by elements of the dietetics and management sector in New Zealand, who currently do not appear to fully appreciate the evolving nature of the field, and the differing validity of various companies offering the tests that form the basis of this personalisation. Various science laboratories are utilising “omics” sciences, including transcriptomics, metabolomics, proteomics and the comprehensive analysis of microbial communities such as the gut microbiota, in order to understand the mechanisms by which certain food products and/or diets relevant to New Zealand, confer a health benefit, and the nature of potential health claims that may be made on the basis of this information. In this article, we give a brief overview of the nutrigenomics landscape in New Zealand since the end of the NuNZ programme, with a particular focus on gastrointestinal health.

## 1. Introduction

It has become increasingly clear that many cases of obesity, cancer, cardiovascular disease (CVD), type 1 and type 2 diabetes, and other chronic diseases that are common in New Zealand, are associated with poor diet and lifestyle (e.g., [1]). However, there is increasing reason to believe that these diseases are also associated with complex interactions between genetic and environmental factors [2].

The term “nutrigenomics” was first proposed by Robert F. Murray, Jr. in 2000 at the 4th International Conference on Nutrition and Fitness, held in Athens, where the author defined it as “the study of the genomic basis for individuality or individual variability in the response to specific nutrients” [3]. The following year it was described as the “new frontier of nutrition science” [4] and was predicted to revolutionise both nutrition research, and its application to consumers through dieticians and nutrition professionals. Kaput and Rodriguez [2] subsequently emphasised how the interface between the nutritional environment and cellular processes (or nutrigenomics) aims to provide a genetic understanding as to how the nutrition of an individual may affect disease susceptibility, by altering the expression or structure of an individual’s genetic makeup. Either an excess or deficiency of certain nutrients or other dietary factors may be important in this respect [5,6,7].

Recognising this important shift in the field of nutrition science, the New Zealand government established a request for proposals to develop the emerging science base of nutrigenomics within New Zealand, and a successful proposal was submitted by what became known as Nutrigenomics New Zealand (NuNZ). 

NuNZ was a collaborative research programme involving AgResearch, Plant & Food Research and the University of Auckland, that ran between 2004 and 2014. As well as developing a nutrigenomics capability in New Zealand, it explored how this capability could be applied for the benefit of the New Zealand food industry. Inflammatory bowel disease (IBD) was selected as an exemplar of a spectrum of diseases, including both Crohn’s disease (CD) and ulcerative colitis (UC), which would benefit from rational tailoring of diet to genotype, in order to minimise malnourishment but also to avoid adverse reactions to certain foods [8,9]. As early as 1996, the results of British twin studies had been published, which emphasised the importance of both genetics and environment in susceptibility to IBD disease development and subsequent progression [10]. Indeed, by 2006, IBD was being seen as an exemplar of gene discovery in complex diseases [11]. The collaborative involvement of academic gastroenterologists, University of Auckland-based Dr Alan Fraser, and University of Otago-based Drs Richard Gearry and Murray Barclay, was essential for the initiation of appropriate studies to better understand gene–diet interactions in IBD in New Zealand. This included using their established clinical links to recruit study participants who identified diet as being an important factor in managing their condition. 

In 2010, NuNZ researchers were fortunate to be invited to join the International IBD Genetics Consortium, which published a subsequently widely-cited paper in 2012 [12]. This confirmed that human genes and diet were important in controlling symptoms of IBD. Subsequent papers emphasised the complexity of IBD risk loci, showing they are enriched in multi-genic regulatory modules encompassing putative causative genes [13,14].

By the end of the NuNZ programme, scientists and clinicians involved in the studies were convinced of the importance of tailoring diet according to genotype. A significant effort went into screening foods commonly eaten in this country, such as feijoas, for potentially beneficial effects. Feijoa extracts were tested in genetically modified cell lines, targeting Toll-like receptor-2, Toll-like receptor 4, and nucleotide-binding oligomerisation domain containing 2 (NOD2), all of which had by then been shown as important genes in IBD susceptibility [15]. Wong and co-workers (2016) pulled together evidence that certain types of dietary fibre were likely to be beneficial through direct interactions with the gut mucosa through immunomodulation, or might act more directly on the microbiome [16]. Thus, while the influence of diet on the effects of genes is still considered to be important, there was increasing recognition around the importance of using omics technologies in order to understand the implications of diet on gene expression [17].

In this review, we will provide an update on the progress of nutrigenetics and nutrigenomics research in New Zealand since the NuNZ programme ended in 2014, including the use of nutrigenetics advice by consumers. We will also briefly describe how some of the omics technologies that were an important part of NuNZ research have continued to develop and be applied to other research areas, in addition to reviewing other technological advances in this area in New Zealand. Finally, we will consider how these technologies might be applied to provide more targeted dietary and nutrition advice, with a particular view to addressing some existing health disparities that are present in New Zealand.

As is shown in Figure 1, since the NuNZ programme ended, the capability developed has continued to grow and contribute to New Zealand’s science landscape, both within the three original partner organisations (AgResearch Limited, Plant & Food Research, and the University of Auckland), and in other NZ research organisations through close collaborations. Three key examples of such collaborating organisations included here are the University of Otago, The Riddet Institute, and Massey University, although there are others. Former NuNZ researchers in each of these organisations continue to make important research contributions to nationally significant NZ science programmes, for example the High-Value Nutrition National Science Challenge (HVN) and the New Zealand Milks Mean More (NZ3M) programme, which is funded by the New Zealand Ministry of Business, Innovation and Employment (MBIE).

## 2. Nutrigenomics Research in NZ

Since the end of the Nutrigenomics New Zealand collaboration, there have been relatively few reports describing research that has specifically focused on improving health outcomes by tailoring nutritional or dietary advice to underlying genetic differences. To gain some insight into the scale of this research, a search of the NIH’s PubMed site was undertaken for “(nutrigenetics OR nutrigenomics) AND (New Zealand OR NZ)”. This identified 65 references of potential relevance, with the earliest of these published in 2006. When considering only those references published after 2014 (i.e., the end of the NuNZ programme), this number was reduced to 26. Of these, 11 were either reviews, perspectives, or position statements and as such did not report any original data. For example, Andraos et al. discuss the approach of combining nutritional measures, subjective methods, and metabolomics profiles [18], while the possibility of using machine learning approaches (currently applied to precision medicine) for application to precision nutrition is also discussed [19]. 

Of the remaining 15 studies, 5 were not carried out in New Zealand, while the remaining 10 included at least one author from the NuNZ programme. Some of these studies were completed during the NuNZ programme using animal models, for example understanding the interaction between the gut microbiota and intestinal inflammation in the interleukin-10 gene-deficient mouse model of IBD [20]. Others described the outcomes of in vitro assays developed within the NuNZ programme, looking at how foods such as feijoas [21] or food components such as sulforaphane [22] found in cruciferous vegetables might influence molecular pathways linked with IBD. In general, while research specifically relating to nutrigenomics continued after the conclusion of the NuNZ programme, it was almost exclusively undertaken by researchers from that collaboration.

When more widely considering the application of omics technologies to nutrition research, the body of research is significantly larger. Again, when searching PubMed and considering only those papers published after 2014, the following numbers of publications were identified:proteomics and (nz OR New Zealand) and nutr*: 46 resultsmetabolomics and (nz OR New Zealand) and nutr*: 94 resultstranscriptomics and (nz OR New Zealand) and nutr*: 81 resultsgenomics and (nz OR New Zealand) and nutr*: 351 results(microbiome OR microbiota) and (nz OR New Zealand) and nutr*: 270 results

In this case, while researchers from the NuNZ programme have made a significant contribution, there have also been many studies from across the NZ research community. Given the rapid development of omics technologies in general, it is not surprising to see the growing application of such technologies in the field of nutrition over this time period, an application which has enabled novel insights into the complex mechanisms by which foods can influence human health.

One example that will serve to illustrate this is the “COMFORT” cohort.

### The COMFORT Cohort 

The Christchurch IBS cohort to investigate mechanisms for gut relief and improved transit (COMFORT) was a cross-sectional observational case control study, initiated in 2016 as part of High-Value Nutrition (HVN), one of New Zealand’s National Science Challenges. For the study, researchers recruited patients who were attending one of two endoscopy clinics for colonoscopy in Christchurch, New Zealand, and a subgroup of participants from the general population who did not undergo a colonoscopy [23]. The aim of the study was to increase the understanding of the underlying disease mechanisms in functional gastrointestinal disorders, including irritable bowel syndrome (IBS), functional diarrhoea (FD) and functional constipation (FC). Demographic, symptom, psychological, dietary and health data were collected, in addition to breath, faecal, blood and urine samples. The cases and controls were predominantly female, with a mean age of around 54 years. Smoking and alcohol consumption rates were similar across the groups. 

The COMFORT study analyses continue, including the application of a number of omics technologies. These analyses are generating increasing evidence that functional gut disorders (FGDs) such as IBS have a microbial pathogenesis [24]. Many metabolites, such as bile acids, short chain fatty acids, vitamins, amino acids and neurotransmitters, can be modified by diet, and in turn, they can modulate the gut microbiome [25]. 

The COMFORT cohort demonstrates how omics technologies that were in part developed with the NuNZ programme have been extended and applied to understanding gastrointestinal disorders other than IBD, which was the original focus of NuNZ. This research will also be important in providing the New Zealand food industry with an underpinning resource to understand how their products can provide benefits to particular consumer groups.

## 3. Uptake of Nutrigenetics in the General Population

Nutrigenetics and nutrigenomics are increasingly recognised as important for the accuracy of clinical nutrition practice [26]. These authors point to the increasing recognition of an individual’s biochemical characteristics, and how genomic information can guide clinical insights into potential nutrient deficiencies and/or excess consumption. A good example of this is the carotene oxygenase *b,b*-carotene-15,15′monooxygenase (BCMO1) gene, one of the genes involved in the conversion of beta-carotene to retinol, which is the biologically active form [27]. Single nucleotide polymorphisms in this gene are common, and may lead to increased disease risk susceptibility, particularly in populations with low intake of foods rich in the active retinol form [28]. 

Joffe and Herholdt stress the growing number of nutrigenetic testing companies that have been established, but also highlight the differences among these. They note the concern that many are being sold direct to consumers, without health care professionals being involved [26]. In 2020, the American Nutrition Association proposed a formal definition of “personalised nutrition” [29], previously known as nutrigenetics. Other terms that have been used include “precision nutrition”, “individualised nutrition” and “genotype-based dietary advice”. Despite differences in the terminology, there is increasing recognition of the need to target dietary advice, to better reflect individual differences in genetics, biochemistry, metabolism and microbiomes. In 2017, an international consortium that included New Zealand scientists, proposed guidelines by which to evaluate the scientific validity and evidence for genotype-based dietary advice [30]. This has proved a useful starting point for several other groups and companies as a guide for both providing dietary advice, and for interpreting genotype-based data. 

A recent update on personalised nutrition [31] noted several points around the use of genetic information for optimising nutritional advice. These included the need to select appropriate genetic variants, the greater affordability and accessibility of genetic tests (as opposed to more complex omics analyses) for implementing in practice, and the fact that evidence from randomised trials shows that DNA-based personalised advice is more effective at modifying behaviour than more general advice, or even personalised advice without a genetic component. The authors also stressed the importance of recognising differences among the various testing companies, and highlight the growing importance of this market for human health.

In 2020, a group from the Academy of Nutrition and Dietetics, including staff from the University of Auckland, published three review articles, in which they expressed concerns about incorporating genetic testing into nutrition care at present, mainly because the field is still evolving. The consensus statement is summarised as follows: “There is insufficient evidence from randomised controlled trials regarding the effectiveness of incorporating nutrigenetic testing into nutrition counselling or care and reporting dietary or clinical outcomes at present. However, research on the application of nutritional genomics to practice is in its infancy, and registered dietitian nutritionists should keep abreast of ongoing developments through continuing education” [32].

Horne and Vohl are among several authors who have questioned the consensus document on a number of grounds. In particular, they expressed concerns that the analysis included a number of studies that disclosed genotype to participants, but the dietary advice given was not based on these genetic data [33]. In contrast, half of the studies tailored nutrition care according to genotype. Koramanoglu and Nielson [34] also cited these three reviews in stressing that the demand for nutrigenomics testing (NGT) is growing, but it is important that health-care professional competence is carefully assessed in relation to this capability. Indeed, Horne and co-workers proposed the development of a nutrigenomics “caremap” [35].

## 4. Obesity and Related Disorders in New Zealand, and the Importance of Nutrigenomics for Preventing These Disorders

As already noted, the NuNZ programme developed its capabilities with a focus on IBD as an example of a disease where nutrition and genetic information could be applied to provide beneficial outcomes. The COMFORT cohort moved this research forward by focusing on other gastrointestinal disorders such as IBS and functional constipation. However, there are a number of other disorders in New Zealand where the application of nutrigenomics could have significant benefit. 

The 2019/2020 annual update of key results from the New Zealand Ministry of Health showed that the prevalence of obesity among adults over fifteen years of age was 30.9%. The prevalence varied by ethnic group, with Pacific Islanders averaging 63.4%, Māori averaging 47.9%, Caucasians averaging 29.3% and Asian adults 15.9%. In addition, there has been a general increase in obesity levels over the previous 8 years for adults aged between 45 and 64 years of age [1]. After adjusting for age, gender and ethnic differences, adults living in the most socioeconomically deprived areas were 1.8 times as likely to be obese as adults living in the least deprived areas. These data confirm the results of several published studies, which also show a high linkage to gout, type 2 diabetes and cardiovascular disease [36,37,38]. Curiously, a discordant association of the CREBRF rs373863828 A allele with increased BMI and protection against type 2 diabetes has been observed in Māori and Pacific Islanders living in New Zealand [39]. This suggests that there are important differences, potentially driven by underlying genetics, that need to be better understood to ensure the most appropriate dietary and nutritional advice for particular ethnic groups in New Zealand.

A Canadian-based study (the NOW randomised controlled trial) has recently shown significantly enhanced long-term adherence to a nutrigenomics-guided lifestyle intervention as compared with individuals with a population-based lifestyle intervention for weight management [40]. Unfortunately, in New Zealand, there appear to be disparities in understanding the genetic bases of some of the health inequities in population groups [41]. Such disparities need to be addressed to enable any nutrigenomics-guided interventions to succeed in an equitable fashion across the different ethnicities in New Zealand.

A number of studies have used nutrigenomics, and applied omics technologies, to assess food or dietary supplements in terms of their value in preventing human diseases.

One dietary intervention study using a dietary supplement was permitted in CD patients, as compared with normal subjects. This tested whether a capsule containing a mixture of long chain omega-3 polyunsaturated fatty acids and vitamin D could enhance the serum levels of both nutrients in either population, after a two-week dietary intervention [42]. The study was justified because of a considerable amount of background work on these nutrients and their beneficial effects on reducing inflammation (e.g., [43]). Nevertheless, the intervention was only permitted for two weeks, and this proved too short a time to show the effects on inflammatory biomarkers, although it did confirm that circulating levels of these nutrients had increased.

Although human intervention studies could not ethically be applied to new food products in patients with IBD, animal models were available. However, such models could not cover all eventualities, and the application of omics technologies in humans could provide an understanding of what was happening. These have been applied to various new food products, for example an olive leaf extract [44] and a novel red apple cultivar [45]. The latter provides a useful example of how omics technologies can be applied to a relatively small number of human subjects over a relatively short time scale. In the study, twenty-five healthy subjects consumed dried daily portions of either a novel red-fleshed or a placebo apple for two weeks, followed by a one-week washout, and then the other type of apple for two weeks. Analysis of the faecal microbiota and of gene expression in peripheral mononuclear blood cells, provided evidence that the anthocyanin-rich apples could positively enhance immune function compared to the control apples, with changes potentially associated with differences in the faecal microbiota. This, along with the outcomes of the COMFORT cohort study, demonstrates that these methods are being applied in a number of ways that are potentially relevant to the New Zealand food industry.

Many findings of human intervention studies, especially in relation to IBD, have been summarised in a review by Laing and co-workers, who were able to consider a personalised dietary approach, thus providing a way forward to manage nutrient deficiencies and food intolerances in IBD [46]. This included both avoiding some potentially adverse effects of diet (in particular a “Western” diet) as well as establishing a tailored approach to providing protection against IBD using diet.

## 5. Importance of the Microbiome

At the start of the NuNZ programme, there was a focus on using omics techniques to assess the interactions between foods or food components and host responses, in particular, changes in intestinal gene and protein expression using transcriptomics and proteomics approaches, respectively. However, a key finding of the IBD genetics consortium research was that the intestinal microbiome had a major role to play in the symptoms of IBD [12]. As this understanding of the important role that the resident microbes within the GI tract play developed, microbiota analysis was increasingly included as an element of any studies. 

The idea that bacteria are involved in IBD was not new at that time. As early as 1960, a review was published regarding the role of bacteria in the pathogenesis of “idiopathic” ulcerative colitis [47]. In 1978, it had been reported that high numbers of E. coli antibodies were present in patients with both UC and CD, and it was suggested that this could play a role in disease perpetuation [48]. Similarly, CD patients were reported to have a statistically significant increase in antibody titre to M. paratuberculosis compared to healthy controls [49]. In the 1990s, it was proposed that in the relapse of IBD there is a breakdown of tolerance to the normal commensal flora of the gut [50], whereas by the early 2000s it was accepted that the intestinal microflora play an important role in the development of IBD [51].

Consistent with this, in the early 2000s, research from the University of Otago provided important insights into the key role of the gut microbiome in human inflammatory conditions [52,53]. In his 2002 publication, Professor Tannock emphasised that the human intestinal microflora is a complex bacterial community, in which obligatory anaerobic species predominate, and which is confined to the distal small bowel and large bowel. He estimated that perhaps 400 bacterial species can occupy the human intestinal tract, but that 30–40 species appeared to dominate. At this time, many of the methods available to categorise the composition of the gut microbiome relied on traditional methods of culture, microscopy and determination of the fermentative and other biochemical properties of bacterial isolates. Subsequent sequence-based technologies have enabled a much greater understanding of the resident microbial population in the GI tract. Nevertheless, a key observation of this publication, namely, the growing evidence that dysbiosis of the intestinal microflora was associated with inflammatory conditions of the human bowel and spine, still holds. As well as Crohn’s disease and ulcerative colitis (UC), he associated such a situation with a tendency to ankylosing spondylitis, and inflammatory disease of the axial, and to a lesser extent, the peripheral joints. A subsequent review by Professor Tannock [54] further emphasised the importance of the microbiome in IBD.

In 2005, Bibiloni and co-authors, including Dr Tannock, tested a mixture of bacteria (probiotics) as treatment for relatively mild UC [55]. They were able to report a remission rate of approximately 77%, with no adverse effects. Additionally, in 2005, both in New Zealand [56] and elsewhere [57,58], research was suggesting that certain dietary fibres could be used as “prebiotics” in order to stimulate the growth of specific gut microbial populations. Such dietary fibres could potentially have beneficial effects in reducing the incidence of colorectal cancer in this country. However, this research also emphasised the need for more sophisticated technologies for studying the microbiome.

One example of the application of microbiome analysis in the NuNZ programme was a study that utilised transcriptomics technologies, and considered the effects of a 6-week dietary intervention with a Mediterranean-inspired diet on inflammation in CD patients [59]. The rationale for such an intervention was based on the evidence that such a diet generally had beneficial effects in susceptible populations (e.g., [60]). The benefit of using transcriptomics was in showing that such a diet not only reduced the established biomarkers of inflammation, but also resulted in altered gene expression, demonstrating the potential impact of diet on the activity of human genes. In addition to the transcriptomics data, this study provided evidence of a trend of the Mediterranean diet to normalise the microbiome, further evidence of the important role played by the microbiome in gene–diet interactions.

In 2017, researchers from New Zealand and Australia reviewed current knowledge on the gut microbiota, claiming that it should be considered as the “new frontier for inflammatory and infectious diseases” [61]. They pointed to the increased understanding of the nature of microbes, including bacteria, fungi, archaea, viruses and protozoa, which occupy the human gastrointestinal tract, and how these microbes influence host disease susceptibility. Diseases associated with an imbalanced microbiome include not only IBD, but also obesity-related inflammatory diseases, allergic and infectious diseases. These authors also highlighted how the development of new, culture-independent assessment techniques are enabling information to be gathered on the majority of gut microbes, which cannot be easily cultured; this increases the understanding of the role they play in the above disease conditions, and more generally in their interaction with the host.

One example of the practical relevance of the microbiome is the concept of faecal microbiome transfer (FMT). The concept of FMT is not new, with the use of faecal matter for a therapeutic purpose first described by Ge Hong, a traditional Chinese medicine doctor, in the fourth century [62]. In Western medicine, the first use appears to be in a veterinary context by the Italian anatomist Fabricius Aquapendente [63]. FMT has frequently been used for the treatment of recurrent Clostridium difficile infection [63,64], and is increasingly considered as a potential therapy for a wide range of diseases and disorders. These include gastrointestinal disorders such as IBD (both UC [65] and CD [66]) and IBS [67], as well as non-GI diseases including non-alcoholic fatty liver disease [68], metabolic syndrome [69], and diabetes [70]. 

A recent example of FMT from New Zealand is the “Gut Bugs” Trial, a randomised double-blind placebo-controlled trial of gut microbiome transfer for the treatment of obesity in adolescents carried out at the Liggins Institute in Auckland [71]. In 2020, a report on the study results revealed that FMT alone did not lead to significant weight loss at 6 weeks. However, it showed signs of reducing visceral adiposity, and improving overall health [72].

## 6. Conclusions

During its ten years, the Nutrigenomics NZ collaboration established a nutrigenomics capability for New Zealand, as well as providing a framework within which a range of omics technologies (including transcriptomics, proteomics, metabolomics, and analyses of the gut microbiome) could be further developed within the partner research institutions. It also supported the development of a number of PhD students and early career postdoctoral scientists, contributing to the ongoing growth of the New Zealand science workforce.

Since the programme ended, the capability developed has continued to grow and contribute to New Zealand’s science landscape, with NuNZ researchers making important research contributions, for example, within entities such as the High-Value Nutrition National Science Challenge. 

Less positively, research specifically focusing on nutrigenomics or nutrigenetics, and the uptake of nutrigenetics (or personalised nutrition) within the nutrition and dietetics community, and more generally by the wider NZ population has not progressed as much as had been anticipated at the end of the NuNZ programme.

Overall, while the NuNZ programme laid the comprehensive groundwork for the ongoing progress in nutrigenomics (both at the research level, for the health benefit of the NZ population, and for use by NZ industry partners to develop high-value foods with potential benefit for particular consumer groups), this has not been built on as much as had been hoped in the intervening years, and significant opportunity remains in this field.

## Figures and Tables

**Figure 1 nutrients-14-01743-f001:**
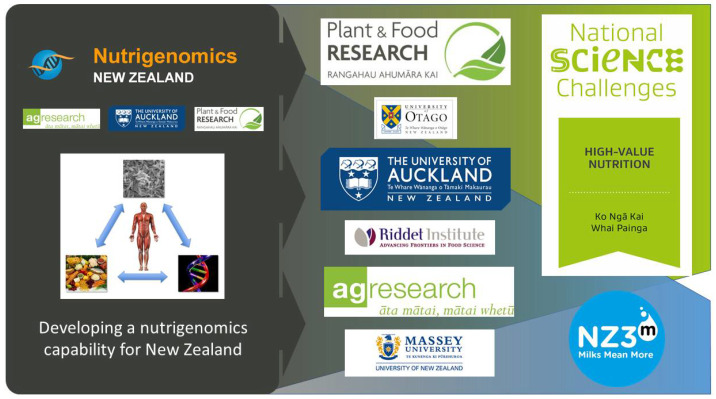
The contribution of the Nutrigenomics New Zealand programme to the NZ research landscape.

## Data Availability

Not applicable.

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
