# Peer review of "Nutrigenomics and Nutrigenetics Research in New Zealand, and Its Relevance and Application to Gastrointestinal Health"

_nutrients, 2022, doi:10.3390/nu14091743_

Round 1

Reviewer 1 Report

Ferguson and Barnett presented the review: “Nutrigenomics and nutrigenetics research in New Zealand” describing the scientific activities conducted by three organizations – The University of Auckland, AgResearch Limited, and Plant & Food Research. The programme ran for the 2004-2014 decade and was aimed at fostering the then novel field of nutrigenomics, in order to assess its potential development in s New Zealand as well as its benefits for the local plant food and agricultural industry.

Interestingly, over the period the programme ran, the Nutrigenomics NZ partnership, not only set up a nutrigenomics capability for New Zealand, but also created a framework within which various omics technologies could be developed through partner research institutions. Thanks to the programme, several academic career paths were set up, both for PhD students and early career postdoctoral scientists. The authors hope that their project will be taken forth in the future years as it is still extremely promising.  

We deeply admire what the Authors’ research effort and commitment have built in New Zealand in the period described in the paper: their article certainly serves the purpose of setting a great example to the global academic community. However, the article does not seem to appropriately fit the genre of a scientific research paper.

 Indeed, while it certainly offers new insights both culturally and historically into the field of NZ nutrigenomics, unfortunately it fails to add novelty to scientific research.

Author Response

Ferguson and Barnett presented the review: “Nutrigenomics and nutrigenetics research in New Zealand” describing the scientific activities conducted by three organizations – The University of Auckland, AgResearch Limited, and Plant & Food Research. The programme ran for the 2004-2014 decade and was aimed at fostering the then novel field of nutrigenomics, in order to assess its potential development in s New Zealand as well as its benefits for the local plant food and agricultural industry.

Interestingly, over the period the programme ran, the Nutrigenomics NZ partnership, not only set up a nutrigenomics capability for New Zealand, but also created a framework within which various omics technologies could be developed through partner research institutions. Thanks to the programme, several academic career paths were set up, both for PhD students and early career postdoctoral scientists. The authors hope that their project will be taken forth in the future years as it is still extremely promising.  

We deeply admire what the Authors’ research effort and commitment have built in New Zealand in the period described in the paper: their article certainly serves the purpose of setting a great example to the global academic community. However, the article does not seem to appropriately fit the genre of a scientific research paper.

Indeed, while it certainly offers new insights both culturally and historically into the field of NZ nutrigenomics, unfortunately it fails to add novelty to scientific research.

Response: We thank the reviewer for their time and effort in reviewing our manuscript, and for their generally positive feedback.

We appreciate that the article does not conform to the requirements of a scientific research paper, and does not add novelty to scientific research. We note that its purpose was as a review to provide the reader with an overview of recent developments in the field of nutrigenomics in New Zealand, with a particular focus on the impact of the Nutrigenomics New Zealand programme. We believe (based on the comments from this and other reviewers) that it has achieved this purpose. We leave it to the journal editors to decide whether this format and purpose are acceptable, and therefore whether the article should be published.

Reviewer 2 Report

The authors mention the Nutrigenomics New Zealand (NuNZ), a collaborative research programme involving three organisations – The University of Auckland, AgResearch Limited and Plant & Food Research- that ran between 2004 and 2014, and explored the nutrigenomics capability in New Zealand and the application for the benefit of the New Zealand food industry and health. Although there is not clear reference to the objective of this article in the abstract, the authors state that they provide an update on the progress of nutrigenetics and nutrigenomics research in New Zealand since the NuNZ programme ended, including the use of nutrigenetics advice by consumers, and the development and use of omics technologies in other research areas.

To obtain the update on the progress of nutrigenetics and nutrigenomics research, as well as the application of omics technologies to nutrition research in New Zealand since the end of NuNZ programme, the authors undertook a search of the NIH’s PubMed site. However, they barely mention the number of articles without including details of the research except for the Christchurch IBS cOhort to investigate Mechanisms FOr gut Relief and improved Transit (COMFORT) study.

Among all the possible diseases in New Zealand, which would benefit from the nutrigenomic approach, and applied omics technologies to assess foods or supplements in preventing human diseases, the authors mention obesity and related disorders. However, the main focus of the report in this aspect is based on the inflammatory bowel disease (IBD), including both Crohn’s disease (CD) and ulcerative colitis (UC), in which the authors demonstrate a relevant role (of their own research). But they leave out many others that are also greatly affected. In addition, the authors mention the importance of the microbiome in the symptoms and development of IBD, with some references for the previous work of the authors. The authors should rethink the title, which is very generic, since the review is limited mainly to IBD diseases.

Although there is no doubt of the possible interest of the review, it is suggested that the authors specify the objective of the manuscript and adapt the title to better reflect the content.

Author Response

The authors mention the Nutrigenomics New Zealand (NuNZ), a collaborative research programme involving three organisations – The University of Auckland, AgResearch Limited and Plant & Food Research- that ran between 2004 and 2014, and explored the nutrigenomics capability in New Zealand and the application for the benefit of the New Zealand food industry and health. Although there is not clear reference to the objective of this article in the abstract, the authors state that they provide an update on the progress of nutrigenetics and nutrigenomics research in New Zealand since the NuNZ programme ended, including the use of nutrigenetics advice by consumers, and the development and use of omics technologies in other research areas.

Response: We thank the reviewer for their constructive and generally positive feedback on our manuscript.

To obtain the update on the progress of nutrigenetics and nutrigenomics research, as well as the application of omics technologies to nutrition research in New Zealand since the end of NuNZ programme, the authors undertook a search of the NIH’s PubMed site. However, they barely mention the number of articles without including details of the research except for the Christchurch IBS cOhort to investigate Mechanisms FOr gut Relief and improved Transit (COMFORT) study.

Response: We take the reviewer’s point that the description of the articles identified by the searches described is overly brief, and we have therefore added more detail regarding the manuscripts identified (in particular those that specifically focus on nutrigenetics or nutrigenomics research, as opposed to the wider application of ‘omics technologies). Please refer to lines 124 to 130 of the revised manuscript.

Among all the possible diseases in New Zealand, which would benefit from the nutrigenomic approach, and applied omics technologies to assess foods or supplements in preventing human diseases, the authors mention obesity and related disorders. However, the main focus of the report in this aspect is based on the inflammatory bowel disease (IBD), including both Crohn’s disease (CD) and ulcerative colitis (UC), in which the authors demonstrate a relevant role (of their own research). But they leave out many others that are also greatly affected. In addition, the authors mention the importance of the microbiome in the symptoms and development of IBD, with some references for the previous work of the authors. The authors should rethink the title, which is very generic, since the review is limited mainly to IBD diseases.

Response: We accept the point that, while we have highlighted obesity and related disorders, there are other diseases of potential interest. We also agree with the reviewer that, with the focus of the NuNZ programme being primarily on IBD, reflecting this in the title is appropriate. We have provided some further (brief) information regarding other potentially relevant diseases, with particular reference to other gastrointestinal disorders highlighted by the COMFORT cohort (lines 168-173). We have also modified the title to reflect the major focus of the review on gastrointestinal health.

Although there is no doubt of the possible interest of the review, it is suggested that the authors specify the objective of the manuscript and adapt the title to better reflect the content.

Response: We accept the reviewer’s comment that the objective could have been more clearly stated, which we have now done at the end of the abstract (lines 25 to 26 in the revised manuscript). We have also revised the title to better reflect this objective.

Reviewer 3 Report

Dear authors,

Your article shares an experience in the field of organizing the research in very important for human health area.

For improvement of your review and attracting of readers I suggest you to add figure that could reflect the impact of Nutrigenomics NZ programme on the future development of the research in different directions.

Author Response

Dear authors,

Your article shares an experience in the field of organizing the research in very important for human health area.

For improvement of your review and attracting of readers I suggest you to add figure that could reflect the impact of Nutrigenomics NZ programme on the future development of the research in different directions.

Response: We thank the reviewer for their positive feedback on our manuscript. As they have suggested, we have included a figure (Figure 1) to show how the Nutrigenomics NZ programme has played a role in future development of the field in NZ, and has contributed to other nationally important research initiatives.

Round 2

Reviewer 1 Report

The Authors improved the manuscript.  They answered to all suggestions